# Exploring the Role of Hsp60 in Alzheimer’s Disease and Type 2 Diabetes: Suggestion for Common Drug Targeting

**DOI:** 10.3390/ijms241512456

**Published:** 2023-08-05

**Authors:** Stefania Zimbone, Maria Carmela Di Rosa, Santina Chiechio, Maria Laura Giuffrida

**Affiliations:** 1Institute of Crystallography, National Research Council (CNR-IC), 95126 Catania, Italy; stefania.zimbone@ic.cnr.it (S.Z.); mariacarmela.dirosa@cogentech.it (M.C.D.R.); 2Cogentech Società Benefit srl Actual Position, 95121 Catania, Italy; 3Department of Drug and Health Sciences, University of Catania, 95125 Catania, Italy; chiechio@unict.it; 4Oasi Research Institute—IRCCS, 94018 Troina, Italy

**Keywords:** Hsp60, Alzheimer’s disease, type 2 diabetes mellitus, IGF-IR

## Abstract

Heat shock protein 60 (Hsp60) is a member of the chaperonin family of heat shock proteins (HSPs), primarily found in the mitochondrial matrix. As a molecular chaperone, Hsp60 plays an essential role in mediating protein folding and assembly, and together with the co-chaperon Hsp10, it is thought to maintain protein homeostasis. Recently, it has been found to localize in non-canonical, extra-mitochondrial sites such as cell membranes or extracellular fluids, particularly in pathological conditions. Starting from its biological function, this review aims to provide a comprehensive understanding of the potential involvement of Hsp60 in Alzheimer’s disease (AD) and Type II Diabetes Mellitus (T2DM), which are known to share impaired key pathways and molecular dysfunctions. Fragmentary data reported in the literature reveal interesting links between the altered expression level or localization of this chaperonin and several disease conditions. The present work offers an overview of the past and more recent knowledge about Hsp60 and its role in the most important cellular processes to shed light on neuronal Hsp60 as a potential common target for both pathologies. The absence of any effective cure for AD patients makes the identification of a new molecular target a promising path by which to move forward in the development of new drugs and/or repositioning of therapies already used for T2DM.

## 1. Introduction

The group of heat shock proteins (HSPs) includes a wide range of proteins the function of which is induced by various physical and chemical stressors, such as hypoxia, exposure to infections, heavy metals, or thermal stimuli, as indicated by the original nomenclature. Despite differences in size, molecular weight, stress inducers, and specific function, HSPs play key roles in the activation of defense mechanisms aimed to maintain protein homeostasis. By virtue of their crucial roles, HSPs are ubiquitous proteins with highly conserved sequences. Hsp60 belongs to group I of the chaperonin family, a structurally related subclass found in organelles of endosymbiotic origin and in the bacterial cytosol [1].Proteins included in this class such as the chloroplast protein complex Rubisco [2], or the GroEL/GroES complex in *Escherichia coli*, are all mainly devoted to assisting the folding of newly synthesized or misfolded proteins and are often responsible for targeting the irreversible unfolded protein for degradation through proteasome or lysosome activity [3]. Hsp60, also known as HSPD1 or Cpn60 in humans, represents the mammalian homolog of the bacterial GroEL. The main intracellular localization of Hsp60 is the mitochondrial matrix, where, together with its co-chaperonin Hsp10, forms a double-barrel structure in which two stacked 7-mer rings bind unfolded polypeptides at its surface, and catalyze the acquisition of proper refolding in an ATP-dependent manner [4].

Under physiological conditions, Hsp60 subunits are encoded by nuclear genes and translated into the cytosol. The primary sequence contains a 26-amino-acids mitochondrial import signal (MIS) that specifically addresses the protein to the organelle. Only after the tag cleavage can Hsp60 reach its final conformation (mtHSP60 or cpn60) through self-assembly, mediated by a pre-existing Hsp60 complex that drives its folding in an ATP-dependent process [5].

Within the mitochondrial matrix, HSP60 plays an important role in the transport and maintenance of mitochondrial proteostasis as well as in the transmission and replication of mitochondrial DNA. The accumulation of denatured proteins or increased ROS production activates the mitochondrial unfolded protein response (UPR^mt^). Similarly to the cytoplasmic ubiquitin–proteasome system (UPS), UPR^mt^ operates a quality control mechanism through the recruitment of chaperones, including Hsp60 and proteases, which promote protein refolding or address damaged proteins to degradation [6]. Cytosolic and mitochondrial proteostasis are tightly connected, as proved by mitochondria-to-nucleus communication (retrograde response), which involves transcriptional activation of Hsp60, and several other proteins involved in the recovery from major organelle stress conditions.

Given its involvement in crucial events related to mitochondrial functioning, Hsp60 holds promise as a potentially interesting marker to discriminate between health and disease conditions. However, from a revision of the current literature, it seems that there are still unexplored aspects of its functions, especially those related to neuronal physiology and pathology.

In this review, we first provide an overview of the newly emerging role of Hsp60 in peripheral cells as well as in the brain. Then, we focus on its involvement in the onset and progression of AD and T2DM, with particular attention given to the molecular mechanisms shared by the two pathologies. Finally, we conclude by highlighting the potential implications of the use of Hsp60 as a common target for the two pathologies.

## 2. Physiopathological Roles of HSP60 in Peripheral and Neuronal Cells

In recent years, the traditional notion that Hsp60 exclusively resides in the matrix has been updated based on new evidence of its presence in non-canonical, extra-mitochondrial sites, where its function seems to be less strictly related to chaperone activity [7,8]. One of the first pieces of evidence of these newly reported localizations referred to cytosolic sites. In a systematic study dating back to 1996, different mammalian cells and tissues were analyzed by immunoelectron microscopy to study the subcellular distribution of Hsp60. Specific monoclonal and polyclonal antibodies raised against Hsp60 were used to better characterize its localization and potential differences among tissues. In all cell types, about 15–20% of the staining was found in extra-mitochondrial sites, such as the cytoplasmic face of the mitochondrial outer membrane, while some foci were identified at the endoplasmic reticulum (ER), vesicles, and plasma membrane [9].

A few years later, the same authors demonstrated an antiapoptotic role for cytosolic Hsp60 in myocytes [10]. They proved that decreased cytosolic Hsp60 leads to an increase in a small unbound fraction of Bax and consequent activation of the apoptotic cascade. More recently, it has been reported that cytosolic Hsp60 directly interacts with the IKK complex, enhancing its activation and promoting, as a downstream effect, cell survival via the NF-κB pathway [11]. A more recent finding has shed light on another interesting pro-survival activity of Hsp60, which was found to form a complex with the smallest member of the inhibitor of the apoptosis protein (IAP) family, survivin [12]. Survivin is highly expressed in several human cancers, showing a positive correlation with tumor progression. Hsp60 and survivin have similar functions and the amounts of these proteins are positively correlated [13,14]. The complex Hsp60-survivin has mainly been investigated in tumor cells, where it was found to reside in both cytoplasm and mitochondria. In these subcellular sites, survivin has been demonstrated to inhibit apoptosis and promote proliferation, making this protein a good target for cancer therapy [14,15,16]. On the other hand, the same complex in the cytosol can participate in cell apoptosis upon death stimuli.

Indeed, a death-promoting effect of cytosolic Hsp60 has been reported in both Jurkat and HeLa cells, where it can accelerate the maturation of caspase-3 by an upstream activator, such as caspase-6, during apoptosis [17]. To explain these apparently contradictory effects, it has been proposed that mitochondrial Hsp60 could be involved in pro-survival/anti-apoptotic functions, while the cytoplasmic fraction could have both positive or negative roles, depending on the status of the protein itself and its source. Cytosolic accumulation of the protein might, in fact, derive from a mitochondrial export, in a way-back mechanism, or from de novo synthesis with an aberrant lacking import to the organelle. Chandra et al. suggested that among the cytoplasmic species of Hsp60, de novo accumulation of the protein might exert a pro-survival function, while the protein released from mitochondria could promote pro-apoptotic functions [18]. More recently, a detailed study of the structure and self-organization of the cytosolic Hsp60 has clarified that the native protein, still bearing the mitochondrial import signal (MIS) that accumulates under specific conditions, is not just an aberrant species that failed to be addressed by mitochondria, but can form functional oligomers capable of playing a physiologic role even in extramitochondrial sites [7,19].

In addition to the abovementioned mitochondrial and cytosolic localizations, in the last few years, although in lower amounts and generally linked to specific conditions, Hsp60 has been also found in other subcellular sites, such as the plasma membrane. Actually, the presence of Hsp60 in the membrane has been known since the late 1990s [20]; however, its role at the cell surface has only recently been clarified. The exposure of Hsp60 on the plasma membrane seems to significantly increase during stress conditions or host infections. Analysis of the binding activity of stressed human umbilical vein endothelial cells (HUVECs) by AFM showed a responsivity of their surface compared to unstressed cells. It has also been shown that Hsp60 is a potent inducer of innate and antigen-specific immunity, representing a good candidate for subunit vaccines or as an adjuvant [21].

Another interesting function, recently described in the N9 microglial cell lines, is the ability of Hsp60 to act as an agonist of the triggering receptor expressed in myeloid cells 2 (TREM-2), an immuno-receptor involved in several diseases for its important role in the inflammation and immune response [22].

The presence of Hsp60 on the plasma membrane has encouraged, in recent years, the investigation of its role as a mediator of cell-cell communication. In 2007, it was proven for the first time that adult cardiac myocytes in both basal and under mild stress conditions release exosomes containing Hsp60, among other components. Exosomal localization of Hsp60 has subsequently been confirmed in tumor cells, where it is highly expressed and secreted, receiving great attention for its potential implications. Indeed, exosomes are emerging as a new type of cancer biomarkers, and Hsp60 seems to be a good candidate for monitoring disease progression and response to treatments, with the great advantage of non-invasive biopsies [23]. The latest data revealed the involvement of exosomal Hsp60 in fibrosarcoma tumor cells [24], human hepatocellular carcinoma cells [25], and human mucoepidermoid bronchial carcinoma, the mechanism of release of which has been also described [26]. Moreover, extracellular Hsp60 has been found not only to be associated with vesicles but also as a free form derived from Golgi transport endosomes [27].

Finally, Hsp60 has also been shown to be present in biological fluids, such as human saliva and plasma serum [28].

Overall, the data reported so far indicate Hsp60 as an interesting marker of disease conditions. We now focus our attention on its involvement in AD and T2DM pathologies.

## 3. Hsp60 Involvement in Alzheimer’s Disease

Despite extensive efforts, continuous failures of clinical trials have been reported in the field of Alzheimer’s disease. Only recently, the approval of immunotherapy drugs by the Food and Drug Administration (FDA) has brought hope for AD therapy [29]. However, the identification of new molecular pathways involved in the onset of the pathology, as well as the search for additional markers of the disease, still remains an urgent need. In this regard, many levels of involvement of Hsp60 in AD progression can be identified, even if most of them are still not completely clarified. The well-known role of Hsp60 as a chaperonin is undoubtedly the first line of evidence for its protective activity in pathologic conditions.

AD is characterized by protein misfolding and the accumulation of amyloid-β and Tau. As for other neurodegenerative “proteinopathies”, proteins that are normally unstructured in healthy brains undergo modifications in their folding, giving rise to small oligomeric and/or large fibrillary aggregates known to be responsible for neuronal toxicity [30]. The molecular basis of protein misfolding has largely been investigated, and although the causes are still under debate, it has been proven that compromised protein accumulation and clearance could play a critical role in neuronal degeneration [31,32]. For this reason, mechanisms able to maintain proteostasis, thus blocking amyloid β1–42 (Aβ1–42) aggregation from the non-pathogenic monomers to toxic soluble oligomers, have been suggested as a promising therapeutic approach in AD pathology [33].

Mangione et al. described the role of Hsp60 in inhibiting the Aβ1–40 amyloid aggregation process even in the absence of ATP and cochaperones [34]. Similarly, in a recent publication, the effects of Hsp60 on Aβ1–42 fibrillogenesis were investigated through several biophysical techniques [35]. The study compared the ability of Hsp60 to counteract Aβ1–42 aggregation with the already-known activity of GroEl to slow down Aβ protofibril and fibril formation [36]. The higher efficacy of Hsp60 to protect neuronal cells from Aβ1−42 toxic insult suggests a gain in functionality through the evolution [35].

A direct interaction between Hsp60 and pre-formed oligomeric species of Aβ1–42 (oAβs) has been further demonstrated by in vitro and ex vivo experiments, showing that Hsp60 treatment was able to: (i) block the oAβs toxic effect in neuroblastoma cells (SH-SY5Y), (ii) significantly reduce the oAβ-driven impairment of long-term potentiation (LTP) and synaptic plasticity in mouse hippocampal brain slices using ex vivo field electrophysiology [37].

Another important level of involvement of Hsp60 in AD pathology is related to mitochondria, which represent its main cellular location and are known to be deeply affected during disease progression.

Hsp60 and Aβ1–42 are normally present in different subcellular compartments; however, pathological conditions such as those in AD might induce protein mislocalization. It is well established that, before the extracellular deposition, Aβ can accumulate in mitochondria [38,39,40,41], contributing to their dysfunction, which is an early event in the pathogenesis of AD [42]. Aβ can be imported into mitochondria via the translocase of the outer membrane (TOM) machinery, independently of the mitochondrial membrane potential [43], while Hsp60, as previously described, may be released from mitochondria to the cytosol where it can reach other cellular compartments or the extracellular space.

Several studies have proposed possible mechanisms by which Aβ might affect mitochondrial function, such as the reduced enzymatic activity of respiratory chain complexes, alterations in mitochondria membrane potential, ATP depletion, and fusion/fission interactions [44,45,46,47].

The presence of the Amyloid-β precursor protein (APP), Aβ, and γ-secretase has been demonstrated in mitochondria harvested from 3xTg-AD mice, a well-established murine model of AD. Proteomics analysis confirmed a strong molecular association between APP/Aβ and Hsp60 in mitochondria from both transgenic and human AD subjects. Moreover, the knockdown of Hsp60 by a viral-mediated shRNA approach proved its role in mitochondrial APP/Aβ translocation [48].

Interestingly, the accumulation of misfolded and/or unfolded proteins in the mitochondrial matrix seems to activate the mitochondrial unfolded protein response (UPR), a transcriptional response that triggers a signaling organelle-nucleus, which in turn, upregulates several key genes involved in mitochondrial proteostasis. In particular, Hsp60, along with other chaperonins, represents one of the multiple mitochondrial UPR markers upregulated in the frontal cortex of subjects who died with sporadic or familial AD [49]. On these bases, defining the role of Hsp60 in mitochondrial UPR may represent a helpful way to promote a compensatory cellular response to be used in AD therapy.

Evidence supporting the above-reported data has also proven the beneficial effect of Hsp60 modulation in AD. Veereshwarayya V. et al. explained the cytoprotective role of Hsp60, alone or in combination with other HSPs, resulting in the rescue of mitochondrial activity in neuroblastoma cells and primary cortical neurons [50]. Using an inducible adenovirus system to raise the levels of intracellular Aβ, the authors showed a specific Hsp60 involvement in protecting vulnerable components of the electron transport chain, such as complex IV, and enzymes of the mitochondrial matrix, but in turn, an inability to interfere with Aβ accumulation or oligomerization.

It is worth noting that the increased expression of Hsp60, Hsp70, and Hsp90, alone or combined, are able to interfere with several stages of mitochondrial apoptotic pathways, defending cells from β-amyloid-induced damage. These data have also been strengthened by the increased immunoreactivity of Hsp60, Hsp70, and Hsp90 observed in rat hippocampus following the injection of Aβ25–35, the most neurotoxic fragment of Aβ in rat hippocampus [51].

Overall, the multiple implications of Hsp60 in AD dysfunctions make this protein an interesting target to be considered.

## 4. Hsp60 Implications in Type 2 Diabetes Mellitus

In recent years, type 2 diabetes mellitus has been studied not only for its pathological features per sé, but also in comorbidity with other relevant disorders, such as Alzheimer’s disease.

Diabetes is one of the most common metabolic diseases in the world. It has been estimated that T2DM accounts for around 90% of all diabetes cases.

Although insulin resistance represents the main feature of the disease, increasing evidence indicates mitochondrial dysfunction and oxidative stress as crucial events taking part in T2DM pathogenesis [52].

In the central nervous system, in particular, the persistent condition of hyperglycemia promotes neuro-inflammation, which is considered to play a key role in the development of vascular dementia, as observed in diabetic patients. Even if the mechanisms are not fully elucidated, a more permeable blood-brain barrier to glucose (influx) seems to be responsible for the overproduction of reactive oxygen species (ROS) and mitochondrial dysfunction.

In this scenario, Hsp60, as a stress-sensitive protein, could represent an interesting key sensor for mitochondrial efficiency and cell viability.

In support of this hypothesis, in vitro studies have revealed an upregulation of Hsp60 and Hsp70 in HeLa cells exposed to a high level of glucose or hydrogen peroxide used to mimic hyperglycemia and oxidative stress conditions, respectively [53].

The upregulation of Hsp60 induced by hyperglycemia has recently been linked to mitochondrial stress, suggesting a molecular connection between diabetes and neuroinflammation [54]. Interestingly, Yuan et al. reported that T2DM patients showed significantly higher levels of Hsp60 in both saliva and serum with respect to non-diabetic controls [28]. The presence of the chaperone in systemic circulation has been proven in healthy and disease conditions [55,56]; however, the biological meaning of this fluctuation and the release mechanisms are not yet fully defined.

The mislocalization of Hsp60 from its canonical site to the plasma membrane or another extracellular localization could represent a cell-to-cell communication mediated by vesicular and/or its free form. Furthermore, it might be reasonable to assume that chaperone translocation could reflect a mitochondrial dysfunction in target cells such as those reported in stressed endothelial cells [57,58] or in cardiac myocytes [59], where Hsp60 has been found on the plasma membrane or secreted by exosomes.

Notably, the ability of Hsp60 to interact with toll-like receptors (TLRs) and activate the inflammatory process, a metabolic characteristic of T2DM, strengthens the idea that this protein could act as a signal in the progression of the disease [60,61].

New insights regarding the role of Hsp60 in the brain have been provided by a recent study carried out on T2DM in vitro and in vivo models. In particular, data showed that in T2DM conditions, hypothalamic insulin resistance and mitochondrial dysfunction were correlated to a downregulation of Hsp60. Interestingly, treatment with leptin was able to restore the mitochondrial activity and integrity in the hypothalamus, increasing Hsp60 levels by a STAT3-dependent pathway. These in vivo results were confirmed in a mouse hypothalamic cell line and in human brain samples from diabetic patients, suggesting that this dysregulation may play a role in the pathophysiology of human T2DM [62].

Several studies have also focused on the analysis of the biochemical and physiological changes observed in diabetic myocardium since cardiovascular diseases typically occur in diabetic patients. The main modifications observed in different forms of cardiomyopathy concern an alteration of glucose and fatty acid metabolism, an increase in apoptotic cells of cardiac muscle cells, oxidative stress, as well as an abnormal expression of growth factors and their receptors [63,64,65]. For this purpose, it has been reported that changes in Hsp60 and IGF-I receptor (IGF-IR) levels specifically occur in the myocardium of diabetic rats, as shown in research performed in streptozotocin (STZ)-induced diabetic rats [66]. The study demonstrated the ability of Hsp60 to modulate cardiac IGF-IR expression. In particular, the observed reduction in cardiac Hsp60 expression seems to contribute to the IGF-IR downregulation in diabetic rat myocardium with a consequent decline in the protective action triggered by the receptor. Further investigations have also been conducted to understand the mechanism by which Hsp60 modulates IGF-1R expression in diabetic rat myocardium and the contribution of the cytoplasmic and mitochondrial pool of Hsp60 to mitochondrial dysfunction and IGF-IR signaling [67,68]. In particular, it has been displayed that in mitochondria-depleted cardiomyocytes, an overexpression of cytosolic Hsp60 induced an increase in IGF-IR expression and signaling by the inhibition of the receptor ubiquitination, suggesting a complex correlation between the chaperon and IGF-1 receptor.

More recently, it was assessed that metformin, a commonly prescribed anti-hyperglycemic agent used in the treatment of T2DM, was also able to counteract ROS production and protein carbonylation through the upregulation of chaperone proteins, such as Hsp60, Hsp70, and LonP1 [69].

## 5. Insulin/IGF Axis and Hsp60: A Common Target for Diabetes and Alzheimer’s Disease

In the last few decades, intriguing studies have revealed several molecular commonalities between AD and T2DM [70]. Brain insulin/IGF resistance and the reduction in glucose consumption found in AD patients have been reported to have a broad range of molecular implications among which are the disruption of signaling pathways that regulate neuronal survival, impairments in energy production, reduced neuronal plasticity, increased activity of kinases that aberrantly phosphorylate tau, together with mitochondria dysfunctions and generation of ROS [71]. All these events negatively affect neuronal survival and activity, promoting neurodegeneration and a decline in cognitive function, which are the main sign of Alzheimer’s disease.

The role of the Ins/IGF axis is crucial in many aspects. Amyloid beta oligomers, which are known to accumulate along AD progression have been reported to interfere with insulin receptor activity [72]; insulin receptor dysfunctions could also be, in turn, responsible for the impairments of oligomer clearance.

Insulin resistance, potentially triggered by oligomers’ chronic exposure, has been recently found in the hippocampal and pre-frontal cortex of post-mortem AD brains, even in patients not affected by diabetes comorbidity. Moreover, it is interesting to note that alterations in brain insulin signaling have been also found in the Down Syndrome (DS) population, which is characterized by early onset of AD-like dementia. In a recent publication, Tramutola et al. demonstrated that markers of brain insulin resistance arise earlier with age in DS compared with the healthy population, and may contribute to the cognitive impairment associated with the early development of AD in patients with DS [73].

Moreover, the contribution of the Ins/IGF axis appears crucial even in physiological conditions. As proven by several recent data, Aβ is not just linked to neuronal toxicity but rather, when normally secreted, it can regulate many key biological functions. Among these, it has been reported to sustain neuronal survival via the phosphorylation of IGF-IR [74], promoting a broad range of IGF-I-like activities such as PI3-K/Akt activation, glucose uptake [75], and CREB-related functions [76].

The downregulation of insulin and IGF receptors observed in the post-mortem brain of AD patients [77] combined with the pauperization of Aβ monomers during oligomers build-up seems to exacerbate the conditions leading to AD pathology.

In this scenario, Hsp60 could represent a key actor, crucially involved in molecular mechanisms shared by AD and Diabetes (Figure 1).

The fragmentary information found in the literature suggests that the protein is sensitive to the micro-environmental changes typical of AD and diabetes, as reported in previous sections and schematically synthesized below and in Table 1:

Hsp60 reported activities:
Protective activity against oAβs toxicity by direct interactionInvolvement in Aβ trafficking from mitochondrial to the cytosolRole as UPR marker in the maintenance of mitochondrial proteostasis

(Protective activity against vulnerable components of the electron transport chain, such as complex IV, and enzymes of the mitochondrial matrix)
iv.Responsiveness to increasing concentrations of glucose (mislocalization, extracellular presence in exosome or biological fluids)v.Modulation of IGF-IR expression in diabetic myocardium

## 6. Hsp60-Based Therapy and Potential Implication in Alzheimer’s Disease and Diabetes 2 Diabetes Mellitus

Among HSPs, in the very recent past, Hsp60 has gained increasing attention in the literature. New functions, not merely related to its chaperonin activity, have been reported, many of which are linked to the extra-mitochondrial localization of the protein.

The increasing knowledge regarding Hsp60 expression, distribution, and functioning has led to the development of new therapeutical strategies with potential implications for several diseases.

One powerful approach to target Hsp60 is the use of chemical inhibitors that include either synthetic and natural compounds. Based on their mechanism of action, the developed inhibitors can be divided into two groups. Type I inhibitors block ATP binding and hydrolysis, thus affecting the refolding activity of Hsp60-Hsp10 [78], while Type II group inhibitors covalently bind Hsp60, interfering with the extra-mitochondrial activities of the protein.

The identification of Hsp60 at the cell membrane or in the extracellular space has also opened up new possibilities for therapeutic intervention. A promising challenge might be tuning by microRNA, cell-to-cell Hsp60-mediated signaling. Mature microRNAs (miRNAs) are small non-coding RNAs that can bind specific sequences at the 3′ UTR of their target mRNAs, interfering with their translation into proteins, thus controlling many biological processes [79].

In a recent publication, it has been proven that the inhibition of miR-1 expression in cardiomyocytes leads to an increased expression of Hsp60, which in turn results in a cardioprotective effect during ischemia [80]. The same microRNA, together with miR-206, was shown to regulate Hsp60 and IGF-1 expression in high glucose conditions, contributing to apoptosis in the reported model of cardiomyocytes [81]. These data might have several implications in AD and T2DM therapy, as the downregulation of cytoplasmic Hsp60 has been shown to contribute to IGF-IR ubiquitination [67] and the failure of insulin and IGF-I signaling is a common feature of diabetes and Alzheimer’s disease.

In fact, even in the absence of comorbidity, an examination of postmortem AD brains revealed reduced IRs and IGF-IR, especially in the hippocampus and hypothalamus, together with increased serine phosphorylation of IRS-1, a key signature of insulin resistance [77].

The inhibition of miR-1/miR-206 could represent a strategy to upregulate Hsp60 and preserve neurons to IGF-IR downregulation.

Recently, an Hsp60-derived peptide was also tested in a silico model as a protective tool against inflammation [82]. Previously published results, dating back to 1996, showed that the Hsp60 peptide p277 was able to arrest autoimmune diabetes induced by the toxin Streptozotocin [83].

In the central nervous system, even if the detailed molecular mechanisms are not still fully clarified, neuroinflammation linked to chronic hyperglycemia conditions seems to contribute to the cognitive decline observed in diabetic patients [84]. As previously described, many studies have provided evidence for mitochondrial involvement in neuron inflammation. Interesting findings have correlated Hsp60 dysfunction and mislocalization to mitochondrial stress, suggesting that it could act as an inflammatory danger signal in the immune responses [85].

Finally, great clinical potential is offered by Hsp60 release in the extracellular space as a free soluble protein or bound to exosomes [86]. Exosomes, in particular, could represent promising candidates for therapeutic interventions in terms of markers of pathologic conditions such as inflammation or molecular targets to be addressed. The measurement of Hsp60, especially in diabetic patients, in which a link with inflammation has been established, can also be used as a novel biomarker of the disease.

## 7. Conclusions

Due to their role in cellular homeostasis as responsive proteins against stressor stimuli, HSPs have been proposed as interesting therapeutic targets to be considered [87], especially in chronic diseases characterized by the accumulation of misfolded proteins, such as AD and T2D [88].

The presence of multiple activities in one polypeptide chain, as reported for Hsp60, mostly related to different cellular localizations, has led to the identification of a new subclass of proteins, named “moonlighting proteins” [89]. As with other “moonlighting proteins”, Hsp60 exhibits more than one physiologically relevant biochemical or biophysical function, an understanding of which could drive the development of novel treatments to be used in therapy.

In this review, we summarized data and information reported so far regarding the contribution of Hsp60 to AD and T2DM. The suggested role of Hsp60, as a common target to address the two pathologies, opens up a new field of pharmacological interventions and/or a potential repositioning of drugs already used for Type 2 Diabetes Mellitus in Alzheimer’s disease, which, to date, is without a resolutive cure.

## Figures and Tables

**Figure 1 ijms-24-12456-f001:**
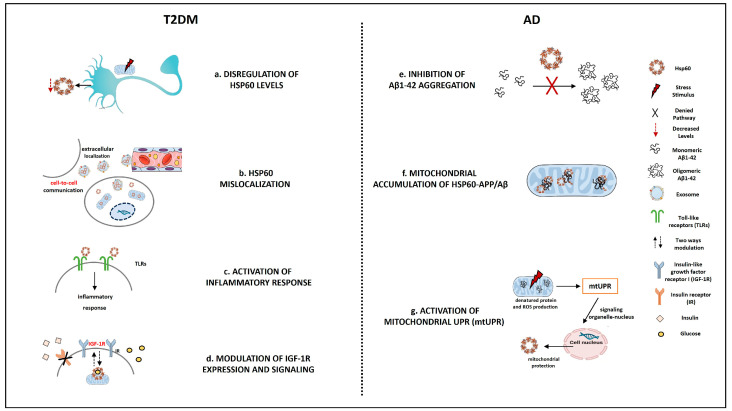
Schematic representation of Hsp60 involvements in AD and T2DM. (**a**) Stress stimuli such as Hyperglycemia, Insulin resistance, or mitochondrial dysfunction induce a dysregulation of Hsp60 levels; (**b**) Hsp60 mislocalizes in pathological conditions such as T2DM; (**c**) Inflammatory responses, typical of T2DM, can be promoted by Hsp60 interaction with TLRs; (**d**) Hsp60 can modulate IGF-1R expression and signaling in diabetic cardiac muscle; (**e**) Hsp60 inhibits pathological aggregation of Aβ1–42; (**f**) Aβ may induce Hsp60-mediated APP/Aβ mislocalization to the mitochondria leading to its dysfunction; (**g**) Hsp60 is upregulated by a retrograde response activated by mitochondrial UPR (mtUPR) in organelle stress conditions. Figure created using the “Mind the Graph” platform.

**Table 1 ijms-24-12456-t001:** Studies reporting Hsp60 functions in Alzheimer’s disease and diabetes.

Pathology	Functions	References
Alzheimer’s Disease	Inhibition of Aβ aggregation	Mangione et al., 2016 [34] Vilasi et al., 2019 [35] Wälti et al., 2018 [36] Marino et al., 2019 [37]
	Mitochondrial proteostasis	Walls et al., 2012 [48] Becks et al., 2016 [49] Veereshwarayya et al., 2006 [50] Ortega et al., 2014 [51] Diaz et al., 2019 [52]
Alzheimer’s Disease/Diabetes	Modulation of Insulin/IGF axis	Chen et al., 2005 [66] Lai et al., 2007 [67] Shan et al., 2003 [68] Docrat et al., 2020 [69]
Diabetes	Cell-to-cell communication	Hall L 2013 [53] Liyanagamage et al., 2020 [54] Yuan et al., 2011 [28] Halcox et al., 2005 [55] Ellins et al., 2008 [56] Cohen-Sfady et al., 2005 [60] Juwono, J et al., 2016 [61]

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
