# Peer review of "Exploring the Role of Hsp60 in Alzheimer’s Disease and Type 2 Diabetes: Suggestion for Common Drug Targeting"

_ijms, 2023, doi:10.3390/ijms241512456_

Round 1

Reviewer 1 Report

The manuscript of the review named: Exploring the role of Hsp60 in Alzheimer’s disease and Type 2  Diabetes: suggestion for common drug targeting, by Zimbone et al, is dedicated to the possible role of molecular chaperonins like Hsp60 with the co-chaperon Hsp10,  which can be found not only in mitochondria matrix but also in neurons, especially in pathological conditions of both Alzheimer disease (AD) and Type II Diabetes Mellitus (T2DM).   Authors suggest potential links between altered expression levels or localization of Hsp60 and various disease conditions and highlighted neuronal Hsp60 as a potential common target for AD and T2DM. This may allow the repositioning of therapies already used for T2DM also to influence AD.

The review is generally well-written and in good English. The author’s work gives an overview of the past and more recent knowledge about Hsp60 and its role in the most important cellular in both pathologies, and this reviewer liked it. We also liked the idea that Hsp60 commonly appeared in additional destinations in pathologies, and remembered that the mislocalization of some proteins is common in some cancers. For example in gliomas organic cation transporters, instead of the external membrane, became mislocalized to the internal membranes and remain functional (Kucheriavykh et al, 2014). Also, it is known that T2DM may be also a protein misfolding disorder because of the aggregation of the islet amyloid polypeptide during this disorder, and the build-up of Hsp60 thus may be of special interest because of its molecular chaperonins functions, similar to AD. Authors can decide whether they will use our suggestions.  

This reviewer endorses that the manuscript is scientifically sound and interesting to a broad group of scientists studying misfolding disorders and is suitable for publication. 

Reviewer 2 Report

In this review (ID: ijms-2504775) entitled “Exploring the role of Hsp60 in Alzheimer’s disease and Type 2 Diabetes: suggestion for common drug targeting”  Zimbone  et al., summarize the literature findings supporting the role of Hsp60 as common terapeutic target for Alzheimer’s Disease and Type 2 Diabetes.

Although this  is an  interesting research topic in the field, in my opinion, this review is not comprehensive in its  present version and some important issues need still to be addressed/completed  prior to be considered for International Journal of Molecular Sciences (IJMS) publication.

Major revisions:

-The introduction is too long and not well-organized. I think that, to make the reading easier,  it should be divided in two different paragraphs : (i) a brief introductive description that highlights the overall aim of reivew and the different biological functions of Hsp60; (ii)  a following paragraph that describes the studies addressing the expression and the phisiopathological  role of HSP60 in peripheral Tissues and in particular in brain .

-The figure 1 is of poor quality. Please, replace it with high-quality image. Besides , a scheme showing the potential molecular mechanisms linking Hsp60 to the onset/ progression of both Alzheimer’s disease (AD) and  Diabetes mellitus (DM) is needed.

-A paragraph referring the development of Hsp60 as potential biomarker for AD and DM is required to make the review exhaustive.

Minor revisions:

-Abbreviations should be clearly indicated in a separate paragraph. 

-The manuscript should be proofread by a native speaker of English.

The manuscript should be proofread by a native speaker of English.

Round 2

Reviewer 2 Report

The authors have addressed all the suggestions of the reviewer.